# Epigenome Wide Association and Stochastic Epigenetic Mutation Analysis on Cord Blood of Preterm Birth

**DOI:** 10.3390/ijms21145044

**Published:** 2020-07-17

**Authors:** Elena Spada, Luciano Calzari, Luigi Corsaro, Teresa Fazia, Monica Mencarelli, Anna Maria Di Blasio, Luisa Bernardinelli, Giulia Zangheri, Michele Vignali, Davide Gentilini

**Affiliations:** 1Department of Brain and Behavioral Sciences, University of Pavia, 27100 Pavia, Italy; elena.spada01@universitadipavia.it (E.S.); luigi.corsaro01@universitadipavia.it (L.C.); teresa.fazia01@ateneopv.it (T.F.); luiber04@unipv.it (L.B.); 2Bioinformatics and Statistical Genomics Unit, Istituto Auxologico Italiano IRCCS, Cusano Milanino, 20095 Milano, Italy; luciano.calza@gmail.com; 3Molecular Biology Laboratory, Istituto Auxologico Italiano IRCCS, Cusano Milanino, 20095 Milano, Italy; m.mencarelli@auxologico.it (M.M.); a.diblasio@auxologico.it (A.M.D.B.); 4Department of Biomedical Science for the Health, University of Milan, Macedonio Melloni Hospital, 20129 Milan, Italy; giuliazangheri@hotmail.com (G.Z.); michele.vignali@unimi.it (M.V.)

**Keywords:** preterm birth, DNA-methylation, genome-wide, Infinium Human Methylation 450K BeadChip, stochastic epigenetic mutations

## Abstract

Preterm birth (PTB) can be defined as the endpoint of a complex process that could be influenced by maternal and environmental factors. Epigenetics recently emerged as an interesting field of investigation since it represents an important mechanism of regulation. This study evaluates epigenetic impact of preterm birth on DNA methylation. Genome-wide DNAm was measured using the Illumina 450K array in cord blood samples obtained from 72 full term and 18 preterm newborns. Lymphocyte composition was calculated based on specific epigenetic markers that are present on the 450k array. Differential methylation analysis was performed both at site and region level; moreover, stochastic epigenetic mutations (SEMs) were also evaluated. The study showed significant differences in blood cell composition between the two groups. Moreover, after multiple testing correction, statistically significant differences in DNA methylation levels emerged between the two groups both at site and region levels. Results obtained were compared to those reported by previous EWAS, leading to a list of more consistent genes associated with PTB. Finally, the SEMs analysis revealed that the burden of SEMs resulted significantly higher in the preterm group. In conclusion, PTB resulted associated to specific epigenetic signatures that involve immune system. Moreover, SEMs analysis revealed an increased epigenetic drift at birth in the preterm group.

## 1. Introduction

Preterm birth (PTB) is defined as birth before 37 weeks of pregnancy. According to the World Health Organization the prevalence of PTB is estimated to be in a range from 5% to 18% in singleton pregnancies [1]. There is a general agreement in considering preterm birth as a syndrome initiated by multiple mechanisms, including infection or inflammation, uteroplacental ischemia or hemorrhage, uterine over distension, stress, and other immunologically mediated processes [2]. Several lines of evidence also supported a genetic predisposition to spontaneous preterm labor and preterm birth [3], while other studies highlighted other maternal risk factors, such as ethnicity, socio-economic conditions and nutritional status during pregnancy have been also reported [2]. Moreover, preterm birth is considered a leading cause of perinatal mortality and long-term morbidity and it is a well-known risk factor for many complex diseases occurring during adult age. Survivors of PTB are more susceptible to several early on-set chronic diseases [4], including obesity [5], cardiovascular disease [6], metabolic disorders [7], respiratory complications [8] and mental and cognitive impairments [9]. Notwithstanding strong epidemiological evidences indicate both genetic and environmental factors as important for PTB, molecular mechanisms surrounding causes and consequences of PTB still remain to be elucidated. Epigenetics is considered a term of connection between genetics and environment and for this reason recently became a natural field of investigation in PTB and DNA methylation, in particular, representing the most studied epigenetic modification. Recent findings in cord blood, identified more than 8800 CpG sites associated with gestational age and the pathway analyses identified enrichment for biological processes critical to embryonic development of brain and lung tissue [10]. Moreover, several studies analyzed cord blood DNA methylation levels comparing preterm and full-term babies and reported a great number of significant differences even if the consistence among these studies has not been evaluated yet [11,12,13,14,15]. Considering the strong background regarding a potential role of epigenetics in PTB, the aim of the present study was to investigate DNA methylation differences associated with PTB at genome wide level and to evaluate consistency of results published so far. Moreover, an innovative analysis able to identify stochastic epigenetic mutations (SEMs) has been also adopted in order to have an estimation of SEMs burden at birth in PTB babies. SEMs have been recently defined as a biomarker of epigenetic drift and of exposure-related accumulation of DNA damage [16] and might have a role in long term effects of PTB.

## 2. Results

### 2.1. Clinical and Immunological Characteristic of Subjects

Clinical and phenotypic traits of pregnancy were examined in order to have an overview of the study population and to detect differences between the two groups analyzed. Results have been shown in Table 1 and summarize the phenotypic characteristics of the 90 subjects recruited for the study, 18 of which are preterm and 72 are full term. Results have been indicated as mean and standard deviation or percentage. A significant difference between preterm and full-term babies emerged considering the weight at birth, the type of delivery (Eutocic delivery or not) and the duration of pregnancy. Furthermore, also blood cell counts, estimated as described in methods section, resulted significantly different between the two groups. The multiple regression model also considered sex and batch effect as potential confounders and indicated that natural killer (NK), Granulocytes and B cells were significantly different between preterm and full-term babies (*p*-value = 0.006, *p*-value = 0.02 and 0.02 respectively). Results have been represented as boxplots in Figure 1. Counts of Lymphocytes CD4T, CD8T, Monocytes, and Plasmablasts were not significantly different between the two groups investigated and results have been represented in Appendix A.

The thick horizontal line represents the median of the distribution, while the box represents the interquartile range. Whiskers are set as the default option for boxplot function and extend to the most extreme data point, which is no more than 1.5 times the interquartile range from the box. Open circles represent outliers (single values exceeding 1.5 interquartile ranges).

### 2.2. DNA Methylation Profiling Using Multi Dimensional Scaling (MDS)

Dimensional reduction was used to visually inspect the dataset for strong signals in the methylation values. The MDS was adopted considering methylation signals from sites and also considering genomic regions: CpG islands, promoters, genes, and tiling. Results have been reported in Figure 2 and show that, considering sites (left panel) and genomic regions (right panels), there are subtle no macroscopic differences in the methylation level between preterm and full-term babies.

### 2.3. Differential Methylation Analysis

Differential methylation analysis was computed both at site and at region level. Blood cells composition, as well as the batch, were used as covariates to adjust the differential methylation analysis. Moreover, a surrogate variables (SV) analysis was also applied in order to correct for potential unmeasured or unmodeled confounders. After FDR adjustment of the *p*-values considering multiple testing, significant differences in methylation levels between cases and controls emerged at site level. The list of probes which resulted differently methylated is reported in Appendix A. Differential methylation analysis performed at regional level was conducted considering CpG islands, promoters, genes, and tiling. Results of the differential methylation analysis at probe and regional levels have been represented by scatterplots and shown in Figure 3. 

### 2.4. Comparative Analysis of Differences in Methylation Status With Previous Epigenomic Wide Association Studies EWAS 

In order to evaluate consistency among studies, results obtained in the present study were compared to those reported by five previous published EWAS that analyzed PTB. This analysis showed that among 2408 significant genes reported by six studies only 389 genes (16.1%) have been confirmed by at least two studies while 99 (4.1%) genes have been confirmed by at least three studies. The upset plot in Figure 4 shows the degree of consistence among studies. Notwithstanding the reduced consistency among studies, two genes, *NCOR2* and *PLCH1*, have been confirmed by all the studies and five genes, *FOXK1, WWTR1, RASA3, PRR5L,* and *IGF2BP1* have been confirmed by five studies. The complete list of these genes and the genomic position involved is shown in Appendix A, while the complete list of genes and studies represented in the upset plot is shown in Appendix A.

### 2.5. Gene Ontology and Functional Analysis

In order to focus the analysis on a more consistent list of markers, the gene ontology classifications analysis was performed considering the list of genes confirmed by at least three studies. The result of the gene ontology analysis has been presented as TreeMap in Appendix A, and shows a significant enrichment in biological processes mainly involved in hematopoietic or lymphoid organ development, negative regulation of transcription and regulation of cell adhesion. 

### 2.6. Stochastic Epigenetic Mutation Analysis

For each subject, the burden of SEMs was calculated as described in the “Methods” section and reported in logarithmic scale. Furthermore, the burden SEMs was also divided into hypo or hypermethylated groups and compared between preterm and full-term groups. The median number of total log(SEMs) was 6.67 (Q1 = 6.0; Q3 = 8.98) in preterm group and 6.0 (Q1 = 5.6; Q3 = 6.5) in the full-term group. Considering the reduced sample size and the unbalanced number of cases and controls, Firth’s logistic regression model was adopted to evaluate differences in the number of SEMs between preterm and full-term babies. The Firth’s logistic regression model, considering sex, blood cellular composition and batch effect as potential confounders, indicated that this difference was significant (*p* = 3.1 × 10^−3^). Moreover, the regression model showed also that both hypo and hypermethylated SEMs resulted in significantly higher in the group of preterm babies (*p* = 3.7 × 10^−3^ and 7.6 × 10^−3^ respectively). The median number of hypomethylated log(SEMs) was 6.3 (Q1 = 5.2; Q3 = 8.83) in preterm group and 5.2 (Q1 = 4.9; Q3 = 5.7) in the full term group while the median number of hypermethylated log(SEMs) was 5.8 (Q1 = 5.5; Q3 = 7.0) in preterm group and 5.2 (Q1 = 5.0; Q3 = 5.7) in the full-term group. Results are shown in Figure 5.

## 3. Discussion

In the present study, we compared cord blood DNAm profiles of preterm and full-term babies. The general consistency among our findings and results reported so far by literature was also evaluated in order to obtain a more robust and consistent list of PTB epigenetic markers. Moreover, we adopted an innovative statistical approach able to identify SEMs and we compared SEM burden between preterm and full-term babies. The key findings from our study included: (i) identification of genes that resulted epigenetically dysregulated in PTB babies in an Italian cohort; (ii) the replication of PTB-associated genes across different studies and ethnicities and identification of a reduced and robust epigenetic signatures of PTB; (iii) the identification of significant differences in blood cell composition between the two groups analyzed; (iv) the identification of a significant increase in SEMs burden at birth in PTB babies. 

The analysis of consistency among previous published studies was important and highlighted a relevant inconsistency among results reported so far. Studies that investigated gestational age in physiological condition were excluded from our analysis in order to reduce heterogeneity. Five studies with similar case control designs met our inclusion criteria and were selected for the analysis. We observed that these studies reported different lists of genes that to a large extent were not consistent among studies. The percentage of completely unconfirmed genes (genes that were reported only by one study) was 50% in the present study, 82.5% in the study by Knijnenburg et al., 71.5% in the study by Cruickshank et al., 64.5% in the study by Febilla et al., 34% in the study by Wu et al., and 47% in the study by Wang et al. The percentage of genes confirmed by at least two studies was only 16.1%. The identification of this inconsistency among results is interesting and reveals the presence of an important heterogeneity. A part of this heterogeneity might be due to statistical issue such as a reduced sample size and statistical power. However, we can speculate that a part of the observed heterogeneity might be due to biological differences and presence of hidden pregnancy factors able to modulate DNA methylation. It is well known for example that a great number of maternal and paternal characteristics, environmental and behavioral exposure factors can significantly modulate the epigenome of the newborn [17]. Moreover, it cannot be excluded that a component of this heterogeneity might also reflect differences in preterm pathophysiology. Notwithstanding the reduced consistency among studies, two genes, *NCOR2* and *PLCH1*, were replicated by all the studies, and five genes, *FOXK1, WWTR1, RASA3, PRR5L,* and *IGF2BP1* were confirmed by at least five studies. This represents an interesting and robust list of epigenetic markers of PTB. The *NCOR2* gene encodes a nuclear receptor co-repressor that mediates transcriptional silencing of certain target genes and it is a co-repressor of glucocorticoid receptor signaling pathways [18]. Interestingly, a recent whole exome sequencing study that investigated PTB reported that rare variants in *NCOR2* could be disease causative [19]. *PLCH1* gene encodes an isoform of phospholipase C and plays a role in inositol signaling pathway. Notwithstanding, this gene has been reported by several studies to be differently methylated in preterm cord blood samples; at this time its role in PTB still remains largely unexplored. However, a recent paper reported that *PLCH1* expression can be modulated during toxoplasma gondii infection [20] and this finding is interesting if we consider that congenital toxoplasmosis can be a cause of preterm birth [21]. *FOXK1* gene is a transcription factor of the forkhead box family (transcription factors that play important roles in regulating the expression of genes involved in cell growth, proliferation, differentiation, and longevity). Hypomethylation and underexpression of *FOXK1* has been recently reported to be associated with intrauterine growth restriction [22]. *WWTR1* gene is a transcriptional co-activator which acts as a regulatory target in the Hippo signaling pathway that plays a pivotal role in organ size control. Recently, it has been reported in a list of genomic biomarkers of prenatal intrauterine inflammation in umbilical cord tissue [23]. *RASA3* gene encodes a protein that binds inositol 1,3,4,5-tetrakisphosphate and stimulates the GTPase activity of Ras p21. Interestingly, recent findings showed a relation among prenatal exposure to air pollutants, alteration in *RASA3* DNA methylation and PTB [24]. Although the exact function of *PRR5L* with respect to pregnancy is unknown, *PRR5L* is a key regulator of cellular *mTORC2* in vitro, which in turn is regulated by lysophosphatidic acid (LPA) activity [25]. *LPA* is implicated in the maintenance of pregnancy [26], uterine contractility [27], and infection-related preterm labor [28]. Finally, *IGF2BP1* encodes a member of the insulin-like growth factor 2 mRNA-binding protein family. There are several studies supporting a role of this gene in PTB describing a role of environment in its epigenetic regulation [29]. Differences between preterm and full-term babies emerged also at immune system levels. NK and B cells counts resulted significantly higher in the preterm group while granulocytes estimates appeared to be significantly lower. This result is not surprising since it has already been described that preterm newborns have immature immune systems, with reduced innate and adaptive immunity [30]. Results reported herein confirmed previous finding showing that preterm infants have a reduced pool of neutrophils, due to reduced granulocyte colony-stimulating factor (*G-CSF*) and granulocyte-macrophage colony-stimulating factor (*GM-CSF*) [30]. Further studies are mandatory to shed light on the consequences for immune development and function of preterm birth. 

The analysis of SEMs represents an innovative approach in the field of PTB and showed a significant increase in the burden of SEMs in the PTB group. SEMs have been recently defined as a biomarker of epigenetic drift and as an indicator of exposure-related accumulation of DNA damage. Moreover, SEMs have been reported to increase exponentially during life and to be associated to several pathological conditions. This result is interesting if considering that a great deal of scientific evidence indicates that survivors of PTB are more susceptible to several early on-set chronic diseases [4], including obesity [5], cardiovascular disease [6], metabolic disorders [7], respiratory complications [8], and mental and cognitive impairments [9]. Molecular mechanisms behind this increased risk have not been clarified yet, however in this study we described for the first time that preterm babies have an increased number of SEMs at birth. This finding might support the hypothesis that the increased burden of epigenetic errors might impair epigenetic regulatory mechanisms and predispose PTB babies to have adverse consequences in adulthood. Furthermore, recent findings described that several other exposure factors like folate supplementation can modulate the number of SEMs in the newborn [17], supporting the idea that it might be possible to control the burden of SEMs at birth thus reducing the risk of long-term adverse effects associated with PTB.

A number of limitations need to be considered when interpreting our results. First, only DNA from cord blood was analyzed. Since cord blood is not necessarily representative of the epigenetic status of all tissues and cells in the newborn, additional studies on other tissues are mandatory to confirm these data. Second, the phenotypic data of the studied subjects did not include all the possible variables related to pregnancy and labor although a surrogate variables analysis was conducted to compensate for this deficiency. Third, although the number of subjects used for comparisons was of the same order of magnitude of previously conducted EWAS, we must underline that a larger sample size could better estimate normal ranges of DNA methylation in each locus. Fourth, the studies selected for the consistency analysis were similar in terms of study design and methodology. We adopted this inclusion criteria in order to reduce heterogeneity among studies; however, we cannot exclude that the list of genes confirmed by the great part of studies might be the result of persistent technical artifacts. Future studies adopting different methodological strategies are mandatory to confirm these findings

## 4. Materials and Methods

### 4.1. Study Design and Study Population

This is a retrospective observational case control study. The control group consisted of *n* = 72 full-term infants (born after the 37th gestational week), while *n* = 18 babies qwerw enrolled in the cases group if born spontaneously before the 37th gestational week. Samples of cord blood from both full-term pregnancies and preterm pregnancies were obtained at the time of delivery by the midwives of the Macedonio Melloni Hospital, Milano, Italy, by puncturing the umbilical vein while the placenta was in utero [31]. The exclusion criteria for both case and control groups were the following: (i) maternal pathologies (nephropathies, essential hypertension, pregnancy-induced hypertension, hypoplasia and uterine malformations, cervical-segmental insufficiency, previous conization, uterine myomatosis, and endometriosis); (ii) uncertain dating of pregnancy; (iii) twinning; (iv) fetal pathologies (fetal malformations and intrauterine growth retardation); (v) fetal appendage pathologies (polyhydramnios, premature rupture of membranes (PROM), placenta previa, and untimely detachment of placenta). Patients were informed that cord blood would be used for research purposes and gave written consent. Approval for this study was been granted by the local Human Institutional Investigation Committee (10122013). Clinical information obtained for each pregnancy included demographic and obstetric factors, pregnancy, delivery, and neonatal outcomes. The sample size was calculated based on the analysis of parameters observed in a previous study in which more than 400 subjects were evaluated [32]. Mean and variance were estimated for all the CpG sites of the array. The number of subjects to be enrolled was calculated assuming an effect size taking into account a difference in the percentage of methylation between groups of at least 10%, by imposing a probability of type I error in the order of 10–7 (level of significance that takes into account the need to correct for multiple testing) and a power of 95%. 

### 4.2. Data Management, Pre-Processing, Normalization and Quality Control

Illumina Methylation 450K raw data were analyzed using the RnBeads 2.0 analysis software package [33]. Sites with overlapping SNPs were firstly removed from the analysis (10119), as well as probes on sex chromosomes (11002). Possible removal of probes and samples of the highest impurity from the dataset was evaluated using the Greedycut algorithm. We considered every β value to be unreliable when its corresponding detection *p*-value was not below the threshold (*t* = 0.05). In order to avoid an erroneous interpretation of stochastic epigenetic variations, probes with coordinates overlapping rare genetic variants annotated in 1000 Genomes and EXAC databases were removed [34]. After the quality control step, none of the samples was excluded for quality reasons while a total of 27,260 probes were removed. The background was subtracted using the methylumi package (method “noob”) [35]. The signal intensity values were normalized using the SWAN normalization method, as implemented in the minfi package. In addition to CpG sites, four sets of genomic regions were covered in the analysis (tiling, genes, promoters, and CpG Islands).

### 4.3. Blood Cell Type Counts

Proportions of CD8 T cells, CD4 T cells, NK cells, B cells, monocytes, and granulocytes were estimated using the “estimateCellCounts” function in the Bioconductor “minfi” package [35], with the reference data for cord blood provided by Bakulski et al. [36].

### 4.4. Differential Methylation Analysis

Differential methylation analysis was conducted both at site and region level according to the sample groups. The *p*-values were computed using the limma method for the site level analysis while a combined *p*-value was calculated from all site *p*-values for the region-based [33]. Regions were defined according to RnBeads definitions [33]:

Genes and promoters: Ensembl (http://www.ensembl.org/index.html) gene definitions were downloaded using the biomaRt package. A promoter was defined as the region spanning 1500 bases upstream and 500 bases downstream of the transcription start site of the corresponding gene.

GpG islands: the CpG island track was downloaded from the dedicated FTP directory of the UCSC Genome Browser (http://genome.ucsc.edu/).

Tiling regions: tiling regions with a window size of 5 kilobases were defined over the whole genome.

In preliminary analyses we adopted different adjustment strategies: (i) without adjustment; (ii) adjustment for batch and cell composition; (iii) adjustment for batch, cell composition and SVA. Results obtained after SVA adjustment were similar to those obtained considering only batch and cell composition as covariates. However, the strategy of also adopting the SVA adjustment was applied based on two main considerationsL (i) based on the consciousness that there are a great number of hidden potential confounding factors in pregnancy that can lead to spurious results; (ii) application of SVA adjustment had no effect on the final list of genes obtained by intersection with previous studies. SV was conducted using the function directly provided in the package RnBeads.

### 4.5. Comparative Analysis with Previous EWAS and Gene Ontology Analysis

We conducted a systematic search of the literature focusing on studies that used a genome-wide approach (Illumina Infinium HumanMethylation27 BeadChip, Illumina Infinium Human Methylation 450K BeadChip and Illumina EPIC 850K) with the aim to compare cord blood methylation levels between full term and preterm babies. In order to reduce heterogeneity, only studies with similar study design and methodology were selected. Studies that investigated DNA methylation modifications associated with gestational age and not with a defined preterm/full term case control study design and studies that evaluated methylation by different experimental approaches were excluded. The five selected studies [11,12,13,14,15] reported differentially methylated genes available in Appendix A. Gene Ontology enrichment analysis has been performed using ToppGene web application (https://toppgene.cchmc.org/help/publications.jsp).

### 4.6. Stochastic Epigenetic Mutation Detection

In order to identify SEMs, we used a method previously described by our group [17,32,37]. Briefly, after the pre-processing step, the distribution and variability of methylation levels were studied in the control populations for all the probes of the array. For each locus, a reference interval for methylation data was calculated considering the formula Q1 − (3 × IQR) and Q3 + (3 × IQR). Thus, for each locus, epigenetic variations were identified as the extreme outliers, with their methylation level lying outside this interval. Finally, all the observed epigenetic mutations were annotated in a new data matrix that allowed to calculate, for each subject, the burden of epigenetic variations and their genomic position. The function adopted to calculate SEMs is published at DOI: 10.5281/zenodo.3813234. In the first step, the analysis described above was applied on control population. Samples were analyzed together and the number of SEMs was calculated in each control subject. In a second step, all the samples in the case group were tested individually and the number of SEMs was calculated for each case subject. 

In order to confirm the power of this analytical approach to detect epigenetic variations, two separate tests were performed on positive controls.

### 4.7. Validation of the SEMs Analysis

Three samples were analyzed in duplicates and epigenetic variations found in each of them were compared. Results showed a mean correlation of 0.99 (*p* < 0.01) among the experiments. The duplicate samples underwent independent bisulfite conversion reaction and this suggests that epigenetic variations are not significantly influenced by bisulfite conversion errors. In the second validation step, 48 whole blood DNA samples obtained from subjects affected by imprinting diseases (Beckwith–Wiedemann syndrome, Angelman syndrome, and Silver Russel syndrome) who underwent diagnostic assays at Istituto Auxologico Italiano were analyzed. For these subjects, a medical report indicating the genomic position of their epigenetic alteration was already available. Briefly, after the identification of the outlier’s probes, a test for over-representation of these probes inside each gene was performed using the hypergeometric cumulative function. The analysis identified genes with enriched number of outlier’s probes (Bonferroni’s corrected *p*-value < 0.05) confirming the presence of the epigenetic alterations previously reported in the medical report.

### 4.8. Statistical Analysis

The “Shapiro.test” function provided in the R package “stats” was applied to test normality among variables. The Bartlett’s test was used to test the homogeneity of variances. The “Wilcox.test” function provided in the R package “class” was used to test differences between cases and controls groups for all non-parametric data. Dimensional reduction was performed using the multiple factor analysis of mixed data approach and the “FAMD” function provided in the R package “FactoMineR”. The univariate and multivariate linear regressions were conducted using generalized linear model and the “glm” function provided in the R “base” package. Considering the reduced sample size and the unbalanced number of cases and controls, Firth’s logistic regression model was adopted to evaluate differences in the number of SEMs and in cord blood cell composition between preterm and full-term babies. FDR correction was performed to adjust for multiple testing.

### 4.9. Data visualization

MDS charts and scatterplots were produced by “ggplot2” and “graphics” packages in R, respectively. The clustering of GO biological processes was visualized using the tool ReviGO (http://revigo.irb.hr/) setting the parameter “Allowed similarity” to Medium (0.7) and referring to the UniProt-to-GO mapping file [38]. Upset plot was produced by “UpSetR” package in R [39]

## 5. Conclusions

In conclusion, in the present study, we compared cord blood DNAm profiles of preterm and full-term babies. We analyzed our results together with previously published findings and obtained a more consistent list of epigenetic markers of PTB. The study described a high heterogeneity among studies and confirmed differences at immune system levels. For the first time, we reported that PTB babies have an increased number of SEMs at birth. This finding might explain why PTB babies have a higher risk of developing a vast array of chronic diseases in adult age.

## Figures and Tables

**Figure 1 ijms-21-05044-f001:**
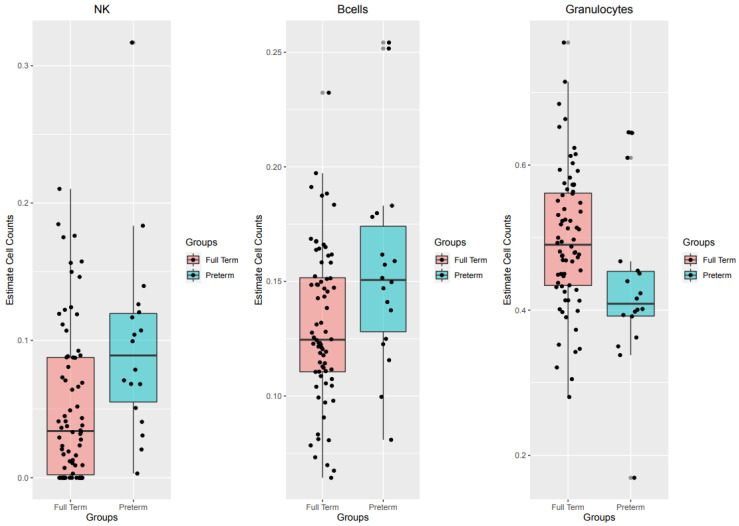
Boxplot showing the distribution of Estimate cell counts between preterm and full-term babies.

**Figure 2 ijms-21-05044-f002:**
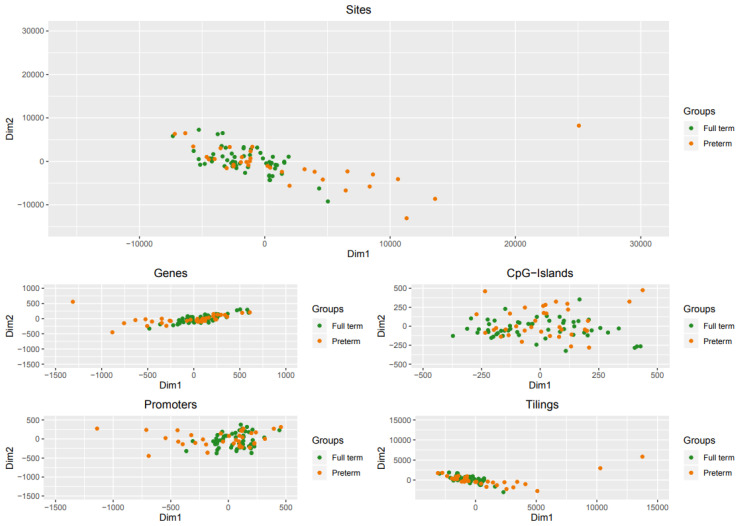
Dimensional reduction of methylation data. Scatter plot showing samples after performing Kruskal’s non-metric multidimensional scaling. Only the first two dimensions are shown.

**Figure 3 ijms-21-05044-f003:**
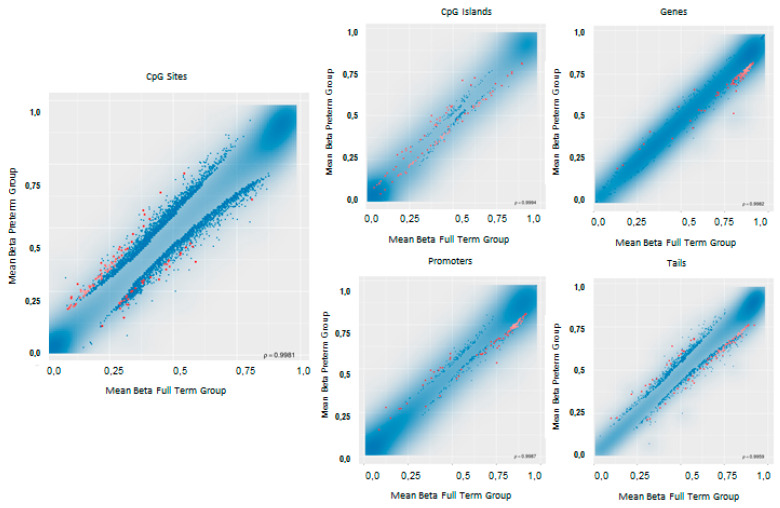
Scatterplot for differential methylation. The transparency corresponds to point density. If the number of points exceeds 2 × 10^6^ then the number of points for density estimation is reduced to that number by random sampling. The 1% of the points in the sparsest populated plot regions are drawn explicitly (up to a maximum of 10,000 points). Additionally, the red dots represent significantly differentially methylated loci (according to the indicated criterion).

**Figure 4 ijms-21-05044-f004:**
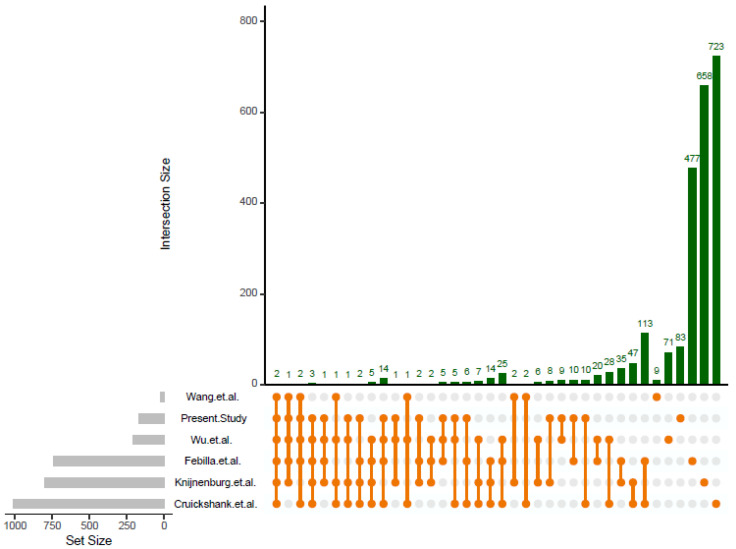
The upset plot showing the degree of consistence among studies.

**Figure 5 ijms-21-05044-f005:**
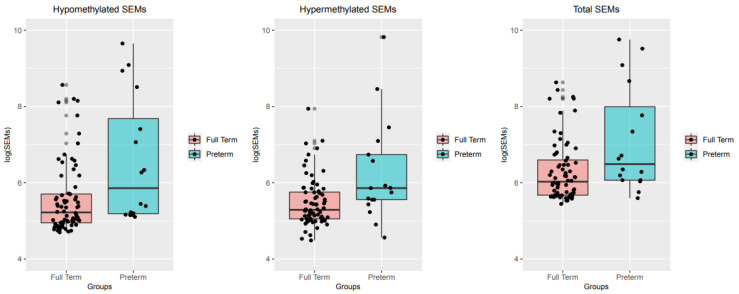
Boxplot showing the distribution of SEMs between preterm and full-term babies. SEMs are divided in Hypomethylated and Hypermethylated and Total. The thick horizontal line represents the median of the distribution while the box represents the interquartile range. Whiskers are set as the default option for boxplot function and extend to the most extreme data point which is no more than 1.5 times the interquartile range from the box. Open circles represent outliers, (single values exceeding 1.5 interquartile ranges).

**Table 1 ijms-21-05044-t001:** Phenotypic characteristics of the 90 subjects recruited: 18 preterm and 72 full-term babies.

Characteristics of Pregnancy	Preterm Babies (*n* = 18)	Full-Term Babies (*n* = 72)	*p*-Value
Sex M	11 (61%)	33 (46%)	ns
Age	33.27 (6.49)	34.77 (5.03)	ns
BMI	21.49 (2.47)	22.42 (3.88)	ns
Weight before pregnancy	56.16 (7.40)	60.40 (12.12)	ns
Increase of weight during pregnancy	10.66 (4.15)	12.22 (4.57)	ns
Diseases during pregnancy	4 (22%)	26 (36%)	ns
Diabetes	2 (11%)	4 (6%)	ns
Hypothyroidism	1 (6%)	10 (14%)	ns
Smoke	4 (22%)	16 (22%)	ns
Assumption of folic acid	17 (94%)	68 (94%)	ns
Pregnancy expressed in days	243.16 (18.81)	275.58 (8.39)	<0.01
Birth weight	2366.11 (520.69)	3251.18(456.15)	<0.01
Eutocic delivery	18 (100%)	53 (74%)	<0.05

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
