# Peer review of "Epigenome Wide Association and Stochastic Epigenetic Mutation Analysis on Cord Blood of Preterm Birth"

_ijms, 2020, doi:10.3390/ijms21145044_

Round 1

Reviewer 1 Report

In this manuscript, Spada et al. performed DNA methylation array analysis of cord blood from normal and preterm birth. The two main findings are: (1) by meta-analysis, the authors find DNA methylation studies of preterm birth to be overall irreproducible, but found a small set of intersecting genes that are mostly consistent between studies, and (2) by SEM analysis, the authors found higher SEM in preterm birth. Before publication, the authors must address a crucial issue for each of these findings: 1. The authors should address the possibility that intersection between multiple studies might be caused by persistent technical artifacts. The authors should discuss how methodology differs between the analyzed studies. 2. The authors should show that SEM analysis is not affected by the imbalance of sample sizes between the case and controls. The observed high SEM in preterm birth could be a technical artifact of their much smaller sample size. Minor points: 1. In Fig 2, scales of the X and Y axes should be equal, because this is MDS. 2. In Fig 3, "Tails" should be "Tiles". The same typo also occurs elsewhere in the main text. 3. In Fig 3 legend, "colored points" should be "red points".

Author Response

We read with interest the Referees' comments and appreciate the time spent to review our manuscript. We were also pleased to read that all the reviewers appreciated the merits of our study,  The comments made were appropriate and constructive. Hence, we modified the manuscript taking into account all the comments and suggestions.

Response to Reviewer 1 Comments

  • The authors should address the possibility that the intersection between multiple studies might be caused by persistent technical artifacts. The authors should discuss how methodology differs between the analyzed studies.

Answer: We better described in the text that studies have been selected to reduce variability and for this reason, we selected studies that adopted the same methodology and study design (illumina platform methylation array, case control studies, preterm vs full term babies). However, we agree with the reviewer that there is the possibility that genes derived from the intersection between multiple studies might be caused by persistent technical artifacts. For this reason, we improved the manuscript highlighting this as a limitation and proposing that other studies adopting different methodologies are mandatory to confirm results. 

  • The authors should show that SEM analysis is not affected by the imbalance of sample sizes between the case and controls. The observed high SEM in preterm birth could be a technical artifact of their much smaller sample size

Answer: We appreciate this comment and we thank the reviewer for his observation. In the first version, based on the robustness of the logistic regression model adopted for the analysis, we didn't consider this issue. However, we agree with the reviewer that the unbalance and the reduced number of samples could lead to biased maximum likelihood estimations. To overcome this problem we adopted an alternative estimation method to reduce this bias. We applied Firth's logistic regression model instead of a classical multiple logistic approach. In the revised version results have been changed based on the new analysis. Moreover, the same method has been also applied for cell composition analysis. The materials and method section has been also modified.

  • In Fig 2, scales of the X and Y axes should be equal, because this is MDS. 

Answer: In the revised version figure has been reproduced as suggested by the reviewer.

  • In Fig 3, "Tails" should be "Tiles". The same typo also occurs elsewhere in the main text. 

Answer: In the revised version this error has been corrected. We thank the reviewer.

  • In Fig 3 legend, "colored points" should be "red points".

Answer: This has been highlighted also by reviewer n 2 and it has been corrected in the text

Reviewer 2 Report

This reviewer thanks the authors for a well written and relatively easy to understand paper. While I do have some suggestions for its improvement, I enjoyed reading it.

Minor points
Though I can guess its meaning, this reviewer is not familiar with the term “trait d’union” , it would be better to use a less colloquial term

Methylation has genome context-specific effects. For the genes found to have differential methylation in multiple studies (such as FOXK1), were the same regions affected (i.e. gene bodies, CpG islands, etc.)?

Comments on figures
Not a major point, but to this reviewer’s eyes, there are subtle differences on MDS, it would be better to say that “Differences between pre- and full-term babies are subtle”. In any case, the conclusions are not affected.

In the Figure 3 legend, did the authors mean to say that the red the colored points represent significantly differentially methylated loci?

The Euler diagram in Figure 4 is quite difficult to parse (though quite pretty). The authors should consider using an UpSet plot instead.

It would be helpful if box plots in Figure 1 and Figure 5 could also show (jittered) data points to help the reader better assess distributions. Alternatively, the authors should use violin plots

In Supplementary Table 3, the “genes from input” links are not working/or are very slow. While listing the GO term’s whole gene list is impractical, listing the “genes from the input” in the supplementary table would help the reader assess relatedness between the enriched categories. Alternatively, a figure (such as one from REVIGO or the cnetplot function from the Bioconductor package clusterProfiler) showing commonalities between the GO terms would be helpful.

Major points
Can the authors comment on the appropriateness of using sva/Combat for batch correction given studies showing the introduction of false positives by this approach (eg. https://www.ncbi.nlm.nih.gov/pmc/articles/PMC7328269/), particularly in unbalanced study designs. How many factors did the authors correct for? What do the results look like without the sva correction (i.e. correcting only for known covariates, batching and cell composition).

Author Response

We read with interest the Referees' comments and appreciate the time spent to review our manuscript. We were also pleased to read that all the reviewers appreciated the merits of our study, the comments made were appropriate and constructive. Hence, we modified the manuscript taking into account all the comments and suggestions.

Response to Reviewer 2 Comments

  • Though I can guess its meaning, this reviewer is not familiar with the term “trait d’union”, it would be better to use a less colloquial term

Answer: The text has been modified has suggested by the reviewer

  • Methylation has genome context-specific effects. For the genes found to have differential methylation in multiple studies (such as FOXK1), were the same regions affected (i.e. gene bodies, CpG islands, etc.)?

Answer: We appreciate this comment and we thank the reviewer. We improved the manuscript with a supplementary table reporting for all study information regarding the probe and the genomic position.

  • Not a major point, but to this reviewer’s eyes, there are subtle differences on MDS, it would be better to say that “Differences between pre- and full-term babies are subtle”. In any case, the conclusions are not affected.

Answer: The sentence has been modified has suggested by the reviewer. We appreciate his suggestion and we recognize that this change improve the realistic description of our results

  • In the Figure 3 legend, did the authors mean to say that the red the colored points represent significantly differentially methylated loci?

Answer:  yes red dots represent the best ranking signals. We modified the sentence has suggested by the reviewer. Reviewer #1 also reported the same error

  • The Euler diagram in Figure 4 is quite difficult to parse (though quite pretty). The authors should consider using an UpSet plot instead.

Answer:  We thank the reviewer for his valuable advice. We changed the figure and we recognized that the upset plot is a better solution to represent the complexity of results.

  • It would be helpful if box plots in Figure 1 and Figure 5 could also show (jittered) data points to help the reader better assess distributions. Alternatively, the authors should use violin plots

Answer:  We thank the reviewer for his suggestion. Figures have been reproduced as suggested

  1. In Supplementary Table 3, the “genes from input” links are not working/or are very slow. While listing the GO term’s whole gene list is impractical, listing the “genes from the input” in the supplementary table would help the reader assess relatedness between the enriched categories. Alternatively, a figure (such as one from REVIGO or the cnetplot function from the Bioconductor package clusterProfiler) showing commonalities between the GO terms would be helpful.

Answer:  We thank the reviewer for his suggestion. Results have been now represented with a supplementary figure. We obtained a treemap graphical representation of GO results with REVIGO. The method section has been also modified.

Major points

Can the authors comment on the appropriateness of using sva/Combat for batch correction given studies showing the introduction of false positives by this approach (eg. https://www.ncbi.nlm.nih.gov/pmc/articles/PMC7328269/), particularly in unbalanced study designs. How many factors did the authors correct for? What do the results look like without the sva correction (i.e. correcting only for known covariates, batching and cell composition).

Answer:  The question highlighted by the reviewer is very important and we agree with him that this step should be considered with extreme caution. In preliminary analyses, we adopted different adjustment strategies. i) without adjustment, ii) adjustment for batch and cell composition; iii) adjustment for batch, cell composition and SVs identified (n=10). We mentioned and described this issue in the revised version of the manuscript. Results obtained after SVA adjustment were similar to those obtained considering only batch and cell composition as covariates. The SVA adjustment resulted to be a little more stringent considering the number of significant signals than the adjustment for cell composition and batch alone. We decided to adopt SVA correction based on two main considerations

1- in the present study the differential methylation analysis was only a preliminary step performed in order to obtain a robust list of genes and signals to be compared with the 5 previous studies. Indeed, the final list derived from the intersection with previous studies and didn’t change depending on the adjustment adopted. However,

2- we previously analyzed and published other results regarding methylation of cord blood and we observed that there are a great number of pregnancy factors that play a role has confounders, moreover literature clearly reported dozens of pregnancy factors that can modulate epigenetics in cord blood and that can be considered potential confounders (folates, BMI of the mother, BMI of the father, age of mother, smoke, diet, stress, air pollution, type of delivery, pathologies, eating disorders, etc). Unfortunately, like other studies, we have no information regarding the great part of these factors that can be considered hidden potential confounders.

In conclusion, we decided to apply also SVA correction based on the consideration that SVA adjustment had no effective effect on the final list of markers obtained by the intersection with previous studies. Moreover, we decided to apply the SVA adjustment because we were conscious that there are a great number of hidden confounding factors that can lead to spurious results (this might be the reason because all studies found different gene sets)

We improved the manuscript adding in the method section a better description of this important step.

Round 2

Reviewer 1 Report

In this revision, the authors have addressed all my concerns. The manuscript is suitable for publication.